# EEG-based vibrotactile evoked brain-computer interfaces system: A systematic review

**Xiuyu Huang[1]\***, **Shuang Liang[2]**, **Zengguang Li[3]**, **Cynthia Yuen Yi Lai[4]**, **Kup-Sze Choi[1]**

**1** Centre for Smart Health, School of Nursing, The Hong Kong Polytechnic University, Hong Kong SAR, China, **2** School of Geographic and Biologic Information, Nanjing University of Posts and Telecommunications, Nanjing, China, **3** School of Computer Science and Technology, Nanjing Tech University, Nanjing, China, **4** Department of Rehabilitation Sciences, The Hong Kong Polytechnic University, Hong Kong SAR, China

\* xiuyu.huang@connect.polyu.hk

**Data Availability Statement:** All relevant data are within the paper and its Supporting Information files.

**Funding:** The study received funding from Hong Kong Research Grants Council (PolyU152006/19E). None of the funding sources impacted the

## Abstract

Recently, a novel electroencephalogram-based brain-computer interface (EVE-BCI) using the vibrotactile stimulus shows great potential for an alternative to other typical motor imagery and visual-based ones. (i) Objective: in this review, crucial aspects of EVE-BCI are extracted from the literature to summarize its key factors, investigate the synthetic evidence of feasibility, and generate recommendations for further studies. (ii) Method: five major databases were searched for relevant publications. Multiple key concepts of EVE-BCI, including data collection, stimulation paradigm, vibrotactile control, EEG signal processing, and reported performance, were derived from each eligible article. We then analyzed these concepts to reach our objective. (iii) Results: (a) seventy-nine studies are eligible for inclusion; (b) EEG data are mostly collected among healthy people with an embodiment of EEG cap in EVE-BCI development; (c) P300 and Steady-State Somatosensory Evoked Potential are the two most popular paradigms; (d) only locations of vibration are heavily explored by previous researchers, while other vibrating factors draw little interest. (e) temporal features of EEG signal are usually extracted and used as the input to linear predictive models for EVE-BCI setup; (f) subject-dependent and offline evaluations remain popular assessments of EVE-BCI performance; (g) accuracies of EVE-BCI are significantly higher than chance levels among different populations. (iv) Significance: we summarize trends and gaps in the current EVE-BCI by identifying influential factors. A comprehensive overview of EVE-BCI can be quickly gained by reading this review. We also provide recommendations for the EVE-BCI design and formulate a checklist for a clear presentation of the research work. They are useful references for researchers to develop a more sophisticated and practical EVE-BCI in future studies.

study design,data collection,analysis,decision to publish,or preparation of this manuscript.

**Competing interests:** The authors have declared that no competing interests exist.

# Introduction

## Overview of brain-computer interface

Over the past decades, the brain-computer interface (BCI) or brain-machine interface (BMI) has developed rapidly with the benefit of advances in brain science and information technology [1]. BCI's primary goal is to enable human and external world interactions by only using brain signals without relying on the brain's normal output pathways, including peripheral nerves and muscles [2]. Many BCI systems have been proposed to offer feasible approaches to interact with the environment for healthy people [1, 3, 4] and patients who partially or entirely lose their motor functions [5–9]. BCI has a promising future for valuable applications with more research focus and the latest innovation.

The BCI system is mostly constructed from consecutive processes, which normally include signal acquisition, data processing, human intention classification, and user feedback provision. The brain activity recording is the basis of the whole series of processes and can be achieved with either invasive or non-invasive techniques [10]. Although the invasive strategy can supply a more precise reading on the brain, it needs surgery to embed the electrodes under the scalp to acquire signals [11]. The side effects caused by surgery may conversely decrease the accuracy of the data collection. On the other hand, the non-invasive BCI collects brain signals by placing sensors on or near the head [12]. Neither surgery nor any painful approaches are needed to implant recording devices. It is much safer and easier to operate than the invasive one [11]. The common non-invasive strategies contain various configurations like the electroencephalogram (EEG) [13], functional magnetic resonance imaging (fMRI) [14], magnetoencephalography (MEG) [15] and functional near-infrared spectroscopy (fNIRS) [16].

## Advantages of EEG and vibrotactile stimuli on BCI

EEG has become the most popular for BCI development among these brain monitoring fashions due to the advantages of economic and portable characteristics [17]. It is an electrophysiological recording approach that measures the brain's electrical variation by electrodes (electrical signal amplifiers) placed on the scalp [13]. Specifically, electrodes collect the aggregation of voltage changes that can arrive at the scalp when pyramidal neurons produce cortex's excitatory postsynaptic potentials [18]. The data collection is more convenient and at a lower cost than other non-invasive acquisition methods, especially fMRI [19]. Also, due to the tremendous electronic propagation speed, the EEG approach can measure electrical changes with milliseconds, which offers an excellent temporal resolution.

Among EEG-based BCIs, the motor imagery (MI) paradigm is one of the most common options. It leads to an event-related desynchronization (ERD) which can be used to detect the user's intention. However, it has two major drawbacks highlighted by researchers. First, motor/movement imagination is an active mental process. Several people may not be able to execute this task. The other one is that ERD patterns of MI require a relatively long time (several seconds) to appear, which limits the practical usage in real-time BCIs. A feasible strategy to improve the information transfer rate (ITR) is to implement faster passive mental events. Event-related potentials (EPRs), such as P300, N100, and N200, are popular choices. These induced paradigms are mainly employed with the visual channel. The primary mechanism is that the visual stimuli elicit distinct brain activity. The EEG signal corresponding to each activity is analyzed to recognize the user's target [20]. Although the visual modality has benefits of intuitiveness and easy control, the usage may be questionable in several cases. For example, the long-time visual focus causes fatigue that affects BCI effectiveness [21]. Moreover, visual

stimulation is not always appropriate for some users, such as patients in the latter stage of amyotrophic lateral sclerosis (ALS), who completely loses visible fasciculations [20]. In this regard, the tactile stimulus may be an interesting alternative. Tactile feedback that stimulates the skin of the surface via direct contact can play the same role as the visual counterpart. The tactile stimuli can be categorized according to the type of sensations: vibration, contact, pressure, temperature, curvature, texture, softness/hardness, and friction [22]. Our review focuses on the BCI using external vibrations, which is more common in the BCI community than other tactile feelings [23]. The vibrotactile sensation performs several edges related to:

◆ easy and precise configuration, where the frequency and amplitude of the vibration can be conveniently and easily customized thanks to the mature mechanical engineering;

◆ distinct pattern, where people can easily detect the difference between vibrating patterns. For example, the difference between 20 and 35 Hz vibration is well-distinguished by humans [24];

◆ widespread usage, although patients with paralysis or ALS may eventually lose motor and visual ability, their somatosensory systems probably remain functioning.

◆ confidentiality, where vibration can be performed by tactors that are covered by clothes.

As a combination of EEG technique and vibrotactile evoked stimulus, EVE-BCI inherits both their advantages. Its components are generally illustrated in Fig 1 (top). Analogous to other evoked BCI systems, external vibrations are first applied to users to elicit distinct brain waves that EEG techniques can capture. Then, the signals are processed by various techniques for feature extraction. Finally, a classifier recognizes the user's intention as an execution command to the outside world. A concrete example of EVE-BCI using the P300 paradigm is

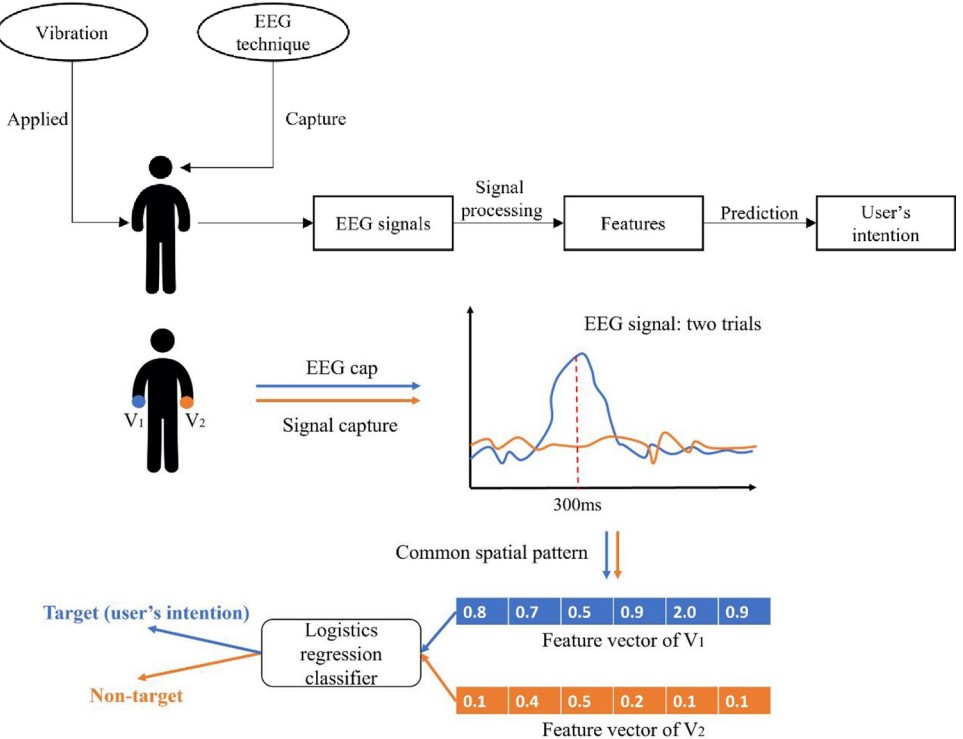

**Fig 1. Illustration of EVE-BCI.** Overall structure (top) and an example of EVE-BCI using P300 paradigm (down).

shown in Fig 1 (bottom). Two vibrators are attached to the subject's left and right index fingers. The subject is asked to pay attention to either finger. Let's assume that he/she focuses on the vibration on the left (target). Left and right tactors keep vibrating in random designated order (the right stimulator vibrates four times as many as the left one at the training phase; two vibrators have the same vibration times at the evaluation phase). An EEG cap records the subject's corresponding brain signals while tactors are vibrating. These signals contain distinct patterns and can be converted into feature vectors by a signal processing method called common spatial pattern (CSP). The logistic regression algorithm predicts the user's focus based on mathematical calculations. It is noticed that there are many settings (e.g., locations of vibrator, signal processing method, and predictive algorithm) in such a simple EVE-BCI. Other choices may contribute to a better design. For example, we can apply the vibrators to wrists instead of fingers. Interesting variables that may advance the performance of EVE-BCI are significant parts of our review.

## Current challenges of EVE-BCI

Although EVE-BCI has such cutting edges mentioned in the previous subsection, it still has some limitations. The EEG, in its nature, contains a few challenges that restrict its effectiveness. First, EEG has a low Signal-to-Noise Ratio (SNR) since the brain waves are usually influenced by the various origins of environmental, physiological, and event-related noise named "artifact" [25, 26]. Second, EEG is a temporal dynamic signal; its characteristic changes across time [27]. In this case, the BCI system generated/trained by the data in a specific period may not adapt to the same user at a different time, raising a significant issue for the long-time EEG device that might require frequent updates. At last, subject dependence is another issue of EEG [28]. This phenomenon appears because of the psychological and physiological variations between people, which results in the difference of voltage magnitude in EEG signals across individuals that significantly affect the BCI system's generality. This individual independence is still a unique challenge in the EEG-based BCI area.

In addition to EEG restrictions, limitations in vibrotactile sensory should also be reminded. The annoyance is one of the drawbacks in applying the vibration on BCI. As the brain event requires to be evoked by constant vibrations during BCI usage, it is easy to result in both physically and mentally uncomfortable feelings. An individual may feel hard to stay focused on the long-time vibration, which dramatically decreases BCI's effectiveness. Another limitation is the inter-subject variability of the vibration frequency. Individuals have various sensitivities on the vibration frequency [24, 29]. For example, someone has the most potent perception on 33 Hz vibration, while another one feels 22 Hz being strongly precepted instead. Testing the optimal vibration frequency for each person costs effort and time, and solutions are not always practical.

## Objectives of the review

Previous research efforts have been devoted to addressing the limitations of EVE-BCI. Sophisticate pre-processing strategies [30, 31], feature extraction approaches [24, 32, 33], and subject-independent classifier [34] were proposed to eliminate the adverse effects of the EEG signal. In the meantime, Shu, Yao [35] tested the difference in EEG patterns between the idle and evoked state and aimed to develop an asynchronous paradigm by using vibrotactile stimulus. An asynchronous BCI enables an individual to use BCI freely without any time slots. It can effectively decrease the exposure of vibration time towards clients and increase the user experience. Also, previous studies tested various vibration characteristics to promote the BCI performance. For instance, multiple positions of vibration such as waist [34, 36, 37], finger [3, 38–

40], and wrist [35, 41] were applied to establish the EVE-BCI. Different values of stimulus onset asynchrony (SOA), also called the total interval of on-time and off-time, were measured in experiments for the optimum BCI system. A fast Fourier transform (FFT) algorithm was applied to detect the human brain's dominant frequency for vibration frequency screening [3, 42, 43]. The screening is used to recognize the individual most sensory frequency. Despite numerous studies focused on EVE-BCI and prove its feasibility, they are presented in various ways and have multiple levels of evidence. No summarization nor synthesis was made to conclude innovations concerning the EVE-BCI. In response to this gap, the present study systematically reviews previous publications about EVE-BCI and aims to offer a comprehensive overview of the current system. This review covers the development trends and the state-of-art in the EVE-BCI by summarizing enormous relevant published articles. We first introduce the general information about the source of publications. The selected studies are then reviewed to identify the following critical concepts for EVE-BCI: data collection, stimulation paradigm, vibrotactile control, EEG signal processing, and reported performance. Lastly, we perform a statistical analysis to explore the feasibility of EVE-BCI across different populations.

## Materials and methods

The systematic review is carried out according to the Reporting Items for Systematic Review and Meta-Analysis (PRISMA) 2020 checklist (see S1 Checklist). The protocol was registered in International Prospective Register of Systematic Reviews (PROSPERO) in April 2021 (CRD42021226760).

### Search strategy and selection

An in-depth search was carried out on Pubmed (Medline), Embase, IEEE Xplore, Web of Science, and PsycINFO. The concept map and the searching terms used are shown in Table 1. Pubmed and IEEE Xplore are indexed by MeSH terms, Embase by Emtree terms, and PsycINFO by Thesaurus terms. Free text search was conducted on all five databases. The searching queries specified for each database are displayed in S1 Appendix. They are formulated based on combinations of three concepts: electroencephalography, brain-computer interface, and vibration.

The studies selected in the review were based on the following criteria: (i) Studies were published between 1st January 2000 to 5th April 2022; (ii) Studies were full-text published in English; (iii) Studies were original research; (iv) Studies were only carried out with

**Table 1. Concept map.**

| Concept | MeSH | Emtree | Thesaurus | Free text |
|---------|------|--------|-----------|-----------|
| Electroencephalography | Electroencephalography | Electroencephalography | Electroencephalography | electroencephalography, electroencephalographies, EEG, eeg, brain electrical activity, electric encephalogram, electro encephalogram, electroencephalogram, brain wave, biofeedback, neurofeedback, neurobiofeedback |
| Brain-computer interface | Brain-Computer Interfaces | Noninvasive brain-Computer interface | None[a] | BCI, Brain-machine interface, Brain-machine interface, direct neural interface, noninvasive brain-computer interface, noninvasive brain computer interface |
| Vibration | Vibration | Vibration | Vibration | vibration, vibrational, vibrotactile, vibrations, vibrate, vibrated, vibrates, vibrating, vibrator, vibrators, vibration sense, tactual, sense of touch, touch stimulus, tactility, tactile stimulation, tactile, touch, haptic |

[a]None: No indexed term for the concept

experiments related to humans; (v) Studies conducted one or more experiments with EEG based non-invasive BCI with only evoked by vibrotactile stimuli.

The searching strategy returned 968 articles with two papers added to the list from the bibliography review (a total of 970). Thirteen articles were first filtered out by the criteria of the publication date and full text of English. The remaining 957 articles were duplicates and properly removed via an approach developed by Bramer, Giustini [44] and double-checked by manual inspection. After the duplicate removal, by reading the title and abstract, 443 additional papers were excluded. Following that, potential eligible studies were assessed in full length. We only included the paper with vibrotactile modality experimental design but excluded the multimodal (even one of them is vibrotactile sensation) or imagery vibrotactile ones. This resulted in a final list of 79 papers in our review. Details of the process are described in the flowchart (Fig 2).

## Data extraction

For research articles finally included in the review, we extracted the following data systematically from each study: (1) sources of studies (including the publication type, publication time, and first author's location); (2) data collection (i.e., target subject and EEG acquisition); (3) stimulation paradigm; (4) vibrotactile control (i.e., frequency and intensity of vibrations, SOA, location of the vibration, and the number of vibrations); (5) EEG signal processing (i.e., preprocessing, feature engineering, and classification); (6) reported performance (i.e., performance metrics and evaluation setting).

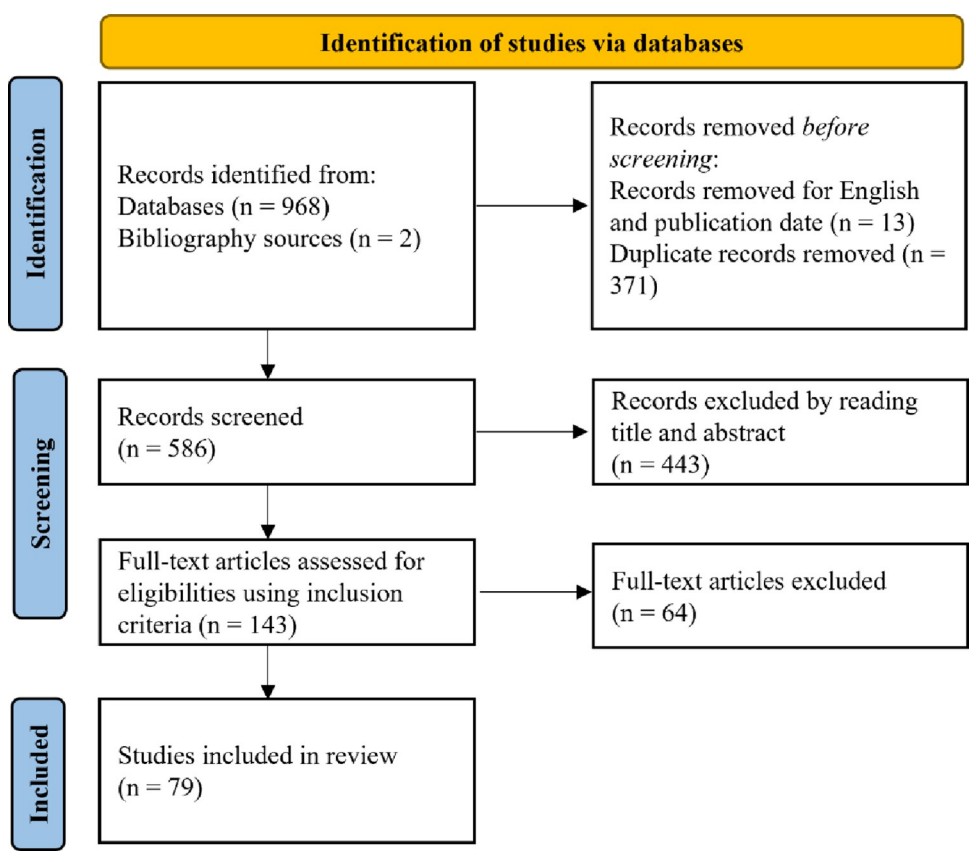

**Fig 2. Flowchart of the selection process.**

## Statistical analysis

The EVE-BCI is drawing attention from the BCI research community. However, compared to the visual-based and motor imagery BCI, its feasibility is not clear. Hence, one of our review's primary aims is to figure out the overall performance of EVE-BCIs on prediction to the user's intention.

To investigate the EVE-BCI feasibility, non-parametric statistical tests were operated to compare the accuracy obtained from publications to the corresponding chance levels. To make the analyses more meaningful, we only included a specific subset of articles:

♦ We selected the study that reports the exact number of the accuracy in its texts or tables. Studies that have ambiguous classification accuracies shown on figures without exact numbers were excluded.

♦ Several studies contained both offline and online evaluations. The online evaluation performance with the highest corrected accuracy [45] is used as the representative for this study since the online evaluation is more realistic than the offline one.

♦ If the study only contains either multiple offline or online evaluations, the one returned the highest corrected accuracy was used.

Three Wilcoxon signed-rank tests [46] were performed to see the difference between accuracies and chance levels of EVE-BCI across healthy subjects, patients, and the overall population. For the overall population analysis, the performances of EVE-BCI for healthy subjects and patients are both considered. Several studies may contain different EVE-BCI experiments for healthy subjects and patients, respectively. They were counted two times (one for healthy and the other for patients) in the statistical test for the overall population.

# Results

## Sources of studies

Our search approach returned 37 journal articles, 39 conference/workshop papers, and 3 book chapters eligible for our inclusion criteria [1, 3, 4, 8, 9, 24, 30–43, 47–105]. These papers were published from 2006 to 2021. Although our search filter includes papers since 2000, no papers were found on this topic before 2006. To the best of our knowledge, the EVE-BCI has not been investigated until Muller developed the first steady-state somatosensory evoked potential (SSSEP) BCI in 2006. Since then, research interests in this field have increased progressively (Fig 3). The most significant number (N = 12) of relevant articles was published in 2018. We also identified these studies' geographical distribution (Fig 4) by looking at the first author's affiliation location. Countries in wealthy regions, including North America, western Europe, and eastern Asia, made the most contributions to this field. We also found that the greatest number of publications were from China (both N = 15).

## Data collection

**Target subject.** EVE-BCIs have been applied to various types of subjects. The BCI system is one of the most promising and popular techniques to assist communication and movement for disabled people. Surprisingly, only around one-fifth of EVE-BCI studies were carried out for patients. These patients suffered from multiple levels of motor functioning loses caused by different diseases including the unresponsive wakefulness syndrome (UWS) [8, 79], disorder of consciousness (DOC) [9, 49, 61, 63, 70], amyotrophic lateral sclerosis (ALS) [54, 57, 87, 89, 103], locked-in syndrome (LIS) [58, 63, 70, 78], stroke [94], and minimally conscious state

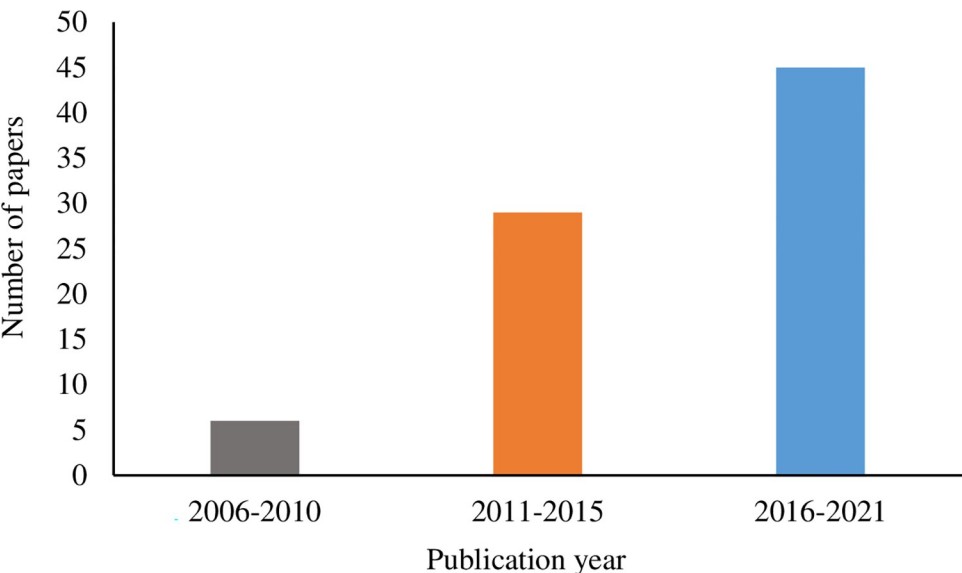

**Fig 3. Publication year of articles.**

(MCS) [86]. On the other hand, most studies (83.54% of cases) alternatively focused on healthy subjects covering adults of all ages, including young people, the middle-aged, and elders. A few studies (5.06%), however, did not report the characteristics of subjects. It is noted that the sum of the above percentage is over 100% because several papers concurrently investigate two or over two types of subjects. A few of them compared the performance of EVE-BCI between healthy controls and patients but returned a contradictory result. A study indicated no difference in the effectiveness of EVE-BCI between these two groups [54], while other two publications [58, 78] point out that EVE-BCI worked better among healthy people. The number of papers regarding various kinds of participants is shown in Table 2.

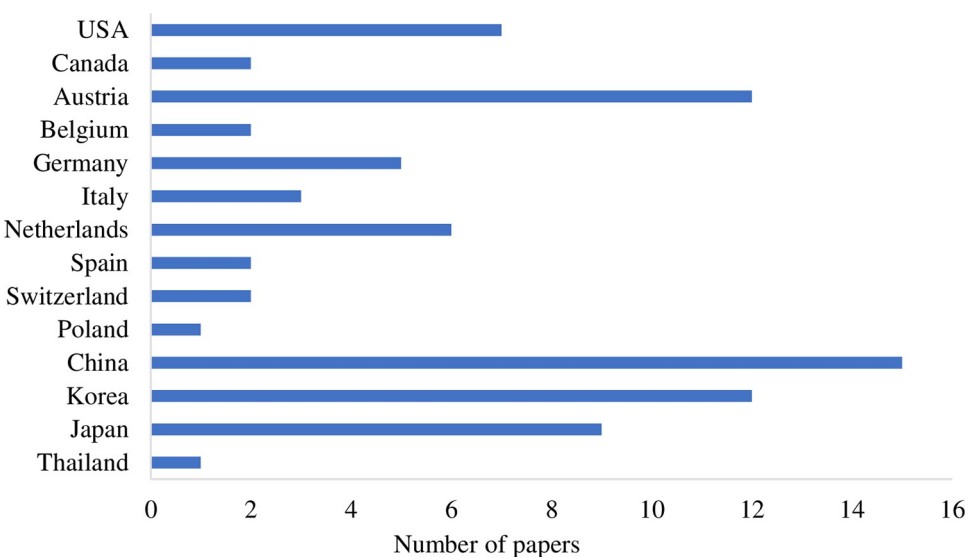

**Fig 4. Geographical distribution of articles.**

**Table 2. Type of targeted subject in EVE-BCI.**

| Subject type | Paper number |
|---|---|
| Healthy | 66 |
| Unresponsive wakefulness syndrome (UWS) | 2 |
| Disorder of consciousness (DOC) | 6 |
| Amyotrophic Lateral Sclerosis (ALS) | 5 |
| Locked-in syndrome (LIS) | 4 |
| Stroke | 1 |
| Minimally conscious state (MCS). | 1 |
| n/a[a] | 4 |
| Two-type[b] | 8 |
| Three-type[c] | 1 |

[a] "n/a" refers to the paper that did not report the type of subject.

[b] "Two-type" refers to the paper that recruited two types of subjects.

[c] "Three-type" refers to the paper that recruited three types of subjects.

Studies also differed in terms of the number of subjects, which determines the reliability of the result. Given subjects recruited in the study are representatives of the population, a larger sample size offers a more reliable outcome with greater precision and estimated power. More than half of the studies were in the range of 2 to 10 (N = 43, 54.43%). Around 30.14% (N = 23) of the articles instead recruited a larger pool of participants, e.g., 11 to 20. Studies involving the above 20 participants contributed 7.59% of cases. The largest sample size among selected papers was 52, with forty DOC patients and twelve healthy people [49]. Only a tiny proportion represented the finding of a case study involving a single subject only, accounting for 5.06% of cases. Nevertheless, three papers (3.80%) did not describe the number of subjects enrolled in their experiments.

**EEG acquisition.** Most EEG acquisition devices used in reviewed publications were in the form of a cap, which assists in the electrode placement on the scalp. It ensures that electrodes are positioned accurately and have enough contact with the head. The cap style was used in all our included articles. It is the most popular style to arrange electrodes in BCI development [106]. It is relatively comfortable to wear for a short period and sufficient for most experimental BCI designs [106, 107]. However, to fix the electrode precisely, the cap is usually very tight, so it often causes headaches with long-term usage [106]. In response to this gap, Blum, Emkes [64] alternatively applied a device called fEEGrid made of pre-attached adhesive foam stickers fitted on the forehead in their EVE-BCI experiments. The electrode grids were affixed directly on the skin without squeeze pressures from the cap. It was regarded to be more comfortable and designated for long-term EEG collection. However, the classification performance of EVE-BCI based on the data collected from fEEGrid (AUC = 0.66) was worse than that from EEG cap (AUC = 0.86), although Blum, Emkes [64] did not report a significant difference.

The spatial densities of EEG devices used in reviewed studies are comparably consistent. No publications have been discovered to use a high-density EEG device (128 channels or over). Seventy-five out of 79 published EVE-BCIs were developed from EEG signals recorded from 64 or fewer channels. The remaining four [82, 83, 93, 99] did not report their EEG equipment's spatial density. Although EEG devices' spatial characteristics were harmonious, channels eventually selected for the actual analysis show significant variations (Fig 5). The number of selected channels was often arbitrarily defined in publications. It was not easy to summarize the exact reasons for their choices. Researchers may be limited by the EEG device available to

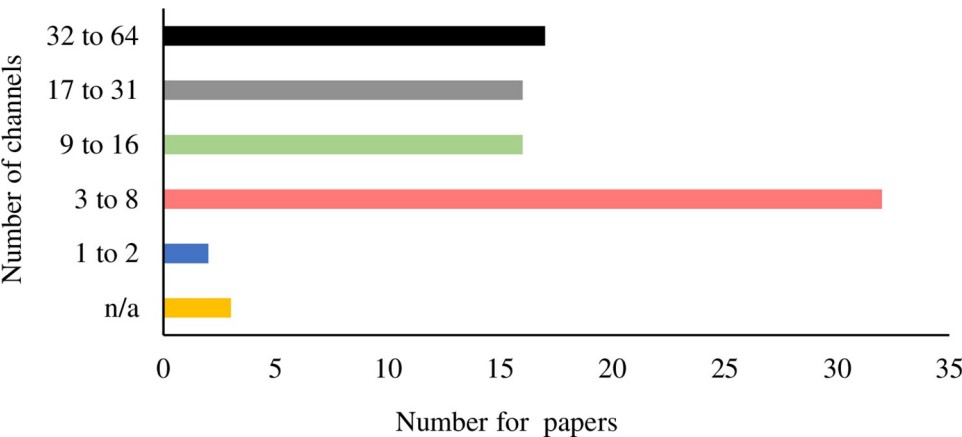

**Fig 5. Number of channels used in EVE-BCI development.**

themselves. Articles in the review suggest that the most frequently used number of channels ranged from 3 to 8. Among these channels, C3, Cz and C4 draw significant interest from researchers. They cover the primary sensorimotor cortex sensitive to somatosensory stimulation and record corresponding neural responses. The first EVE-BCI was developed using EEG data collected in this region. Since then, over 92% (73 out of 79) of studies used the data from C3, Cz and C4 in their BCI experiments. The rest, only a tiny portion, either only used data from other channels or did not report any relevant details. The patterns of brain events were also visualized in reviewed articles. Twenty-eight reviewed articles selected specific channels, instead of all, for the visualization. More than 85% of these publications chose at least either one of the C3, Cz and C4 channels to present the patterns. Those results suggest the intensive interest of the primary sensorimotor cortex region in the EVE-BCI development.

## Stimulation paradigm

The present review focuses on the BCI utilizing the brain signal elicited by the external vibrotactile stimulus. One of the most frequent paradigms employed was event-related potential (ERP). The BCIs using EPRs can be performed without prior training and do not present the "BCI illiteracy" issue. It is a proposed aspect wherein users fail to obtain a certain level of performance in using a BCI during a standard training process [108]. Another major strength of ERP deployment is its reproducibility. The ERP can be evoked stably across different people and different periods [109]. P300 was the most commonly used ERP and performed in 46 out of 79 among our reviewed studies. To our best knowledge, the first ERP-based EVE-BCI using the P300 paradigm was established by Aloise, Lasorsa [74] in 2007. Their study compared three modalities (e.g., visual, audio, and vibrotactile stimulus) for P300-based BCI. Although the vibrotactile-based one had the worst performance, it still achieved an average of 68% accuracy, significantly higher than the chance level (12.5%). This finding proved the feasibility of the vibrotactile-based P300 paradigm. Other ERP paradigms (Table 3) including N100 [60], N200 [54, 60] and error-related potential [48, 53, 62] were also performed in the EVE-BCI. However, only one, two, and three studies investigated them, respectively. In [60], Thurlings et al. compared the performance of EVE-BCI using different ERP paradigms. No significant difference in performance was found when using the features of N100, N200, P300, or a combination of them.

 Another popular vibrotactile paradigm in BCIs was SSSEP. It composes a resonant steady-state component of brain signals elicited by successive vibration within 17–35 Hz frequency

Table 3. Types of event-related potential in EVE-BCI.

| Paradigm | Description |
|---|---|
| P300 | The P300 is a positive deflection in voltage of EEG signal with a latency of 300ms after an unexpected stimulus. It is often elicited by the oddball paradigm and occurs on the parietal lobe. |
| N100 | The N100 refers to an immense, negative-going evoked potential that occurs around 100 milliseconds after presenting a stimulus. It can be detected by electroencephalography and distributed mainly at the fronto-central region of the scalp. |
| N200 | The N200 is also an event-related potential measured by EEG. It is a negative-going brain wave that peaks at approximately 200ms after a stimulus's onset and is recognized at the scalp's anterior region. |
| Error-related potential | The error-related potential is elicited by the perception of an error and measured through EEG. It consists of two components: error-related negativity (Ne) and error positivity (Pe). The Ne is a negative potential peaking at 50-100ms post-stimulus. The Pe is a positive potential following the occurrence of Ne. The Pe can be additionally categorized into a frontocentral and a centroparietal component. The frontocentral one arises immediately after the Ne, and the centroparietal one is at peaking around 200–400ms after the error. |

[83]. As stated by Muller [85], who proposed the first SSSEP-based BCI (to our best knowledge), the advantage of this approach was the independence from the visual modality channel [85]. It works as an effective alternative to visual-based P300 or steady-state visual evoked potential (SSVEP) for those with visual impairments. In Muller et al.'s study, two vibrotactile stimuli with different frequencies were applied on both index fingers (frequency of right finger minus frequency of left finger = 5 Hz). The user was asked to focus on one vibration at each time. By analysing the corresponding brain wave elicited by each vibration, the BCI system classified the user's attention on the left or right. More than one-third (n = 29, 36.71% of cases) of our reviewed EVE-BCIs depended on the SSSEP. In addition to SSSEP in a low-frequency range of 17 to 35 Hz, Chen, Fu [3] proposed a novel paradigm called vibrotactile induced Sensory-Motor Rhythms (VT-SMR). The usual Sensory-Motor Rhythm (SMR) refers to the oscillatory event in the somatosensory and motor areas of the brain. It usually appears when performing a motor execution or motor imagery. Chen, Fu [3] tested whether the SMR could also be induced by vibrotactile stimulation without any actual or imagery motor tasks. More specifically, VT-SMR is regarded as an oscillatory idle of synchronized electric brain events aroused by constant vibrotactile stimuli. Unlike the SSSEP, where the vibration frequency is the frequency of interest in EEG, the vibration frequency deployed in VT-SMR was not limited to the effective bandwidth of EEG. Thus, the frequency band of EEG analysed in their study could exceed the vibration frequency.

Instead of exhibiting only one popular paradigm, either P300 ERP or SSSEP, two reviewed studies [30, 31] used two paradigms simultaneously. They both introduced a hybrid BCI, which utilized vibrotactile stimulus to evoke both SSSEP and transient ERP (i.e., P300) at the same time. However, they demonstrated opposite findings. Pokorny, Breitwieser [31] did not report a significant difference in the BCI classification when using the combined features compared to using features of a single paradigm. Alternatively, in [30], the hybrid classifier had a significantly higher classification accuracy than either SSSEP-based or P300-based one.

## Vibrotactile control

The vibrotactile control plays a significant role in the EVE-BCI establishment. The review summarizes the vibration design in the following four aspects: frequency and intensity of vibrations, SOA, location of vibrations, and number of vibrations.

**Frequency and intensity of vibrations.** As stated in the Introduction section, the proper detection for individual sensitivity to the vibration frequency remains a significant challenge

in EVE-BCI. A precise detection can theoretically enhance the performance of the BCI [31, 40]. However, in our reviewed articles, only around 18% of cases implement vibration frequency screening for individuals before the formal EEG collection. All these cases are in the SSSEP paradigm where brain signals' oscillations resonate with the vibration frequency, so observing the individual most robust perception of vibration is a vital step and necessary to be executed. Two screening approaches were used. (1) Task-based method [24, 31, 85]: vibrations in a specific range of frequencies (e.g., 13-35Hz) were subsequently applied to subjects with an increment of 2 Hz. During the vibration, participants were required to solve a mathematical equation and ignore the vibration. The EEG collected in the task period was transformed through FFT into frequency-domain data. The frequency with the highest amplitude was considered to evoke the most distinct EEG pattern and be the resonance-like (optimal) frequency. (2) Response-time-based method [97]: vibrations in various frequencies were also performed on subjects. Instead of addressing the equation, they needed to press the button as quick as possible once they felt the vibration. The frequency that led to the largest response time variation was regarded as the individual optimal one. The remaining studies either did not describe relevant details or solely applied the fixed frequency to all subjects. Several of them [30, 32, 40, 41, 52, 56, 59, 65] were even in the SSSEP design. No initial frequency screening was found in ERP-based EVE-BCIs. Among these studies, the selection of vibration frequency was usually in an arbitrary way without clear rationalities. The applied vibration usually fell at 20 to 40 Hz or above 150 Hz. It was also worth emphasizing that no investigation into the effect of vibration frequency (independent variable) on the performance of the EVE-BCI was found among returned publications.

In addition to the frequency, the intensity of vibrators is also explored in this review. Only around 16% (n = 13) of articles reported their choices of intensity ranging from 0.6–4.0 G. They normally stated that participants have explicit feelings about the vibration, but no special or sophisticated designs were applied. Furthermore, most reviewed publications do not present any details about the vibrotactile intensity.

**Stimulus onset asynchrony (SOA).**   Analogous to the vibration frequency, only few publications (two cases) [57, 91] explored how the SOA affected the performance of EVE-BCI. In [91], vibrations were applied to multiple positions around the waist. Oddball context [110] was used to evoke P300 ERPs. The optimal on-time and off-time were found at around 188ms and 63-188ms, respectively, to reach the highest bitrate (bits/min). This setting was similar to that in many visual P300-based BCIs. Additionally, Kaufmann, Holz [57] proposed a case study to investigate a tactile BCI communication system with different SOAs. The offline classification achieves superior accuracy in the condition of short on-time (200ms) and long off-time (1000ms). In other publications, the SOA was usually pre-defined and fixed in the EVE-BCI experiments. The ERP paradigms, including P300, N100, N200, and error-related potential, tended to have shorter SOAs. The stimulus on-times and off-times respectively range from 100 to 600ms and from 0 to 1400ms across different articles. These short-time designs are because ERPs are elicited by transient stimulus (sudden feeling). On the contract, other paradigms (SSSEP and vibrotactile evoked SMR) require a relatively long-time perception [83, 94]. The mechanism behind these paradigms is to detect the resonant oscillation of the brain signals generated by the vibration, so a longer SOA can result in a more obvious resonant activity pattern that benefits classification performance. The on-time and off-time (e.g., 2–11.5s and 2–9.5s) were operated in the second level rather than the millisecond one.

The experiments typically contain many repetitive trials in data collection and evaluation for the BCI system. If SOA were always the same, a subject can usually get used to it and predict when the stimulus will occur. This phenomenon can significantly degrade the reliability of data. An intelligent skill was performed to prevent the SOA adaption. More than 20% (N = 16)

**Table 4. Number of vibrations.**

| Vibration number | 1 | 2 | 3 | 4 | 5 | 6 | 7 | 8 | n/a[a] |
|---|---|---|---|---|---|---|---|---|---|
| Paper number | 2 | 38 | 22 | 8 | 8 | 11 | 1 | 4 | 1 |

[a] "n/a" refers to the paper that did not report the number of vibrations.

of papers used an off-time randomly selected from an interval, such as 4.5–6.5s [35, 72], to increase prediction difficulty. No paper was found to apply the random on-time stimulus to decrease the adaptive effect.

**Location of the vibration.** Unlike the vibration frequency or the SOA, the location of vibration draws a lot of interest from researchers in the EVE-BCI area. Many locations throughout the body including fingers [30, 31, 50, 95], wrist [8, 9], back [37, 48, 58], foot [43, 61], ankle [1, 63], shoulder [76, 88], waist [37, 60, 62], arm [34, 57], leg [34, 88], cheek [1], neck [66, 86], chest [73], elbow [76, 92], hand palms [89], scapula [96] and knee [103] were explored in publications. More than 1/3 (N = 30) of articles carried out the experiments with tactors attached in multiple parts of the body such as the wrist-back [78, 79, 87], finger-toe [24, 42], arm-waist-leg [34, 88]. Multiple-finger attachment or multiple positions on the waist, etc., were not considered as the multiple-part attachment. Other studies only performed tactors on a single part of the body, except one [71] did not report its vibrating position. The most popular spots for the tactor were fingers that account for 30 cases. They were estimated as the most delicate area of sensation [31, 95, 111], where the thumb and index fingers were used in most cases. Following fingers, wrists (N = 25), and waist (N = 14) were also common attempts due to their advantages of both sensitivity and confidentiality (under clothes). Other positions mentioned above, however, respectively constituted less than 5 cases and were not common ones for EVE-BCI development.

**Number of vibrations.** The determination of the vibration number is also an essential component of vibrotactile control. It presented a significant dissimilarity among studies (Table 4). The two-vibration design was built in nearly half of returned studies. Studies with a great number of vibrations (e.g., 6 and 8) were all in the design of the P300 paradigm, where the ERP was evoked by the deviant stimulus among many standard stimuli. Additionally, it is critical to understand that the number of the vibration is not necessarily the same as the number of targets (classes). For example, in a P300 based EVE-BCI developed by [87], eight vibrations on different positions were applied. Seven of them were standard vibrations, while only one was deviant. The subject was asked to focus on the deviant vibration and ignore the rest. The EVE-BCI correspondingly makes the binary classification (2 classes). In this condition, the number of stimuli was larger than the number of targets. This phenomenon always occurred in the ERP-based design. On the contract, the number of stimuli was smaller than the number of targets in several SSSEP designs. For instance, Yao, Chen [72] proposed an EVE-BCI with vibrations on both wrists. In different experimental trials, the subject needed to either concentrate on neither, left, right, or both wrists, resulting in a four-class BCI system that only contained two vibrations. These two scenarios (either larger or smaller) were common in our review studies accounting for 32 cases. The number of vibrations and the number of targets were consistent in the rest of the 47 articles.

## EEG signal processing

The core goal of the EVE-BCI system is to detect users' intentions through analyzing EEG signals. The analysis framework usually comprises pre-processing, feature engineering, and

classification in BCI development. Hence, our review investigates these three steps and summarizes implementations from previous publications.

**Pre-processing.** As EEG has low SNR, pre-processing is usually a preferable step before extracting patterns from EEG. Pre-processing aims to reduce the noise but keep useful information as much as possible. Many pre-processing techniques (Table 5), including bandpass filtering, artifact removal, baseline correction, down-sampling, detrend, logarithm transform, and spatially whiten, were applied in articles returned from the database. We did not find any standard pipelines for the implementation of these techniques. Different combinations were operated in various studies. For example, in [63], no pre-processing was applied to EEG data. Similarly, Nam, Cichocki [32] only used a 0.5 to 40 Hz bandpass filter as the whole pre-processing step. On the contract, a relatively complex pipeline was performed in Annen, Mertel [49]'s study. The 0.01–30 Hz bandpass filter was firstly applied, followed by an artifact and outlier removal where trials were rejected if any data points exceeded the threshold of $\pm100\mu V$. Then, data were further baseline corrected and down-sampled using a sliding-window average approach. Although different studies had their own choice of techniques, most research realized the importance of the pre-processing step. Only six studies neither used any pre-processing methods nor showed the relevant information.

The bandpass filter was the favourite strategy in the pre-processing stage. It is one of the most common signal processing methods and can be automatically done in many EEG handling software. It effectively eliminates the data out of the interesting bandwidth. The 0.1–30 Hz bandpass filtering drew the greatest attention in EVE-BCI [4, 34, 55, 58, 78, 87, 96]. Followed by the bandpass filter, down-sampling was the second prior option, which may be due to the high sample rate of EEG collection devices. The factoring [78, 92], window average [60], and window sum [50] were popular ways to reduce samples of the high dimension EEG data in articles. The artifact removal was also frequently operated. It was used to take away certain kinds of noise, including ocular and muscular artifacts. The removal of artifacts is a key factor achieve high EEG decoding performance [112]. Generally, both automatic and manual ways were used. One of the automatic strategies was to define thresholds, such as $\pm70\mu V$ [66, 89], $\pm90\mu V$ [30], or $\pm100\mu V$ [39, 61, 70], for artifact detection. The EEG trials which contained data points out of the threshold were regarded as bad ones and abandoned. Additionally, independent component analysis (ICA) running by computer was also used to divide ocular components from original EEG signals [40, 59]. Nevertheless, manual inspection was considered a more reliable way to remove artifacts [56, 81]. It is time-consuming and heavily relies on the knowledge of the human expert. Lehne, Ihme [62] used a hybrid method to overcome this issue. They successively operated ICA and visual inspection. Major flawed trials were first rejected by ICA and subsequently by experts. This method achieved a good performance of artifact removal and considerably reduced the human workload.

**Table 5. Types of pre-processing strategies in EVE-BCI.**

| Pre-processing strategy | Paper number |
|---|---|
| Bandpass filtering | 68 |
| Down-sampling | 37 |
| Artifact removal | 27 |
| Baseline correction | 17 |
| Logarithm transform | 4 |
| Detrend | 4 |
| Spatial whitening | 1 |

**Feature engineering.** In the EEG processing framework, feature engineering is a compelling stage for dealing with high-dimension data. It starts from a long sequence of EEG data and derives informative and non-redundant patterns. It can significantly reduce dimensionality, save computational time, and avoid the overfitting issue. Features in the development of the EVE-BCI were usually handcrafted ones extracted using mathematical approaches.

EEG features comprise spatiality, frequency, and temporal. EEG signals' spatiality is most frequently determined by variants of CSP method [41]. It separates multi-channel EEG signals into subcomponents that have maximum distance invariance between two classes [113]. This strategy enhances the spatial dissimilarity between the target and non-target EEG signals. In addition to the standard CSP, two modified CSP algorithms, including filter bank common spatial filtering (FBCSP) [33] and sub-band common spatial filtering (SBCSP) [40], were also used. They both apply multiple bandpass filters before the CSP operation, distinguishing the spatial difference in each specific band. For frequency features, the fast Fourier transform was popularly adopted. For instance, in [24, 43, 90], it converted the time-domain into frequency-domain representations. Frequencies with the largest amplitudes and amplitudes were extracted as features for the classifier. Furthermore, the frequency-domain information regarding the power spectral density (PSD) was also used as a feature in [56, 65]. The average band power accumulated by the PSD in the frequency band of 16 to 31 Hz was assigned to the left-right finger SSSEP pattern recognition [56]. On the other hand, raw EEG data itself is the temporal-domain representation. After down-sampling in the pre-processing step, many reviewed articles used them directly as the feature input for the classification. The principal component analysis (PCA) was the technique found to extract further temporal information that contained the highest factor loading [59, 64, 67]. The distribution of features used in publications is shown in Fig 6.

Most EVE-BCI systems (N = 68, 86.08%) only employed a single type of feature (i.e., spatiality, frequency, or temporal). Others (N = 11, 13.92%) preferred to use a combination of two features. For instance, Nam, Koo [82] used CSP features and the largest amplitude in the

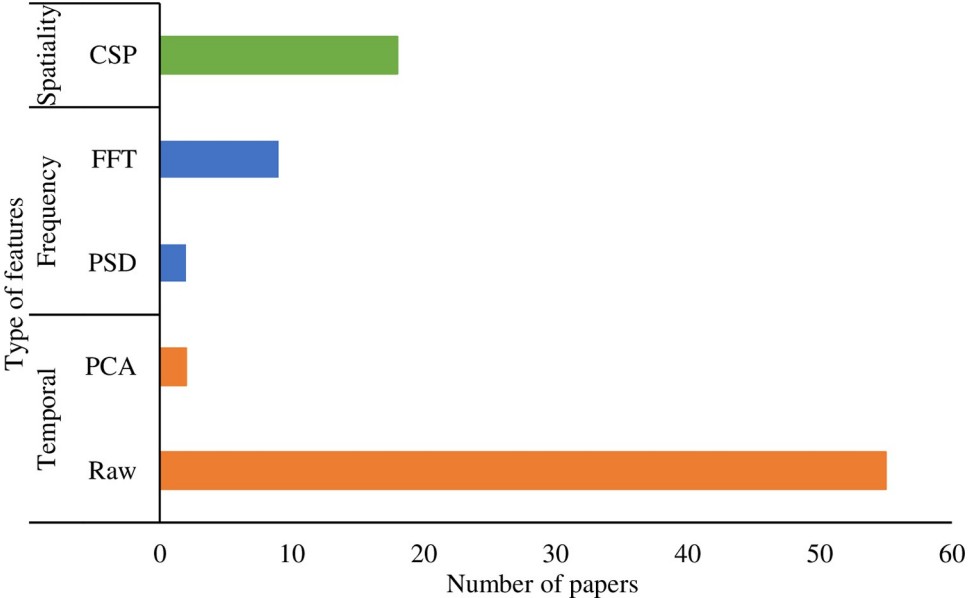

**Fig 6. Types of features used in EVE-BCI development.**

frequency domain as the input for the classification. Yet, no combination of using all three types of the feature was found in existing EVE-BCIs.

**Classification.** In general, EVE-BCI in reviewed articles focused on classifying the user's intention using distinct patterns of certain brain events. Classification is a procedure of establishing a model that can make accurate predictions to unlabelled/unseen samples in a set of categories [114, 115]. Current research efforts in EVE-BCI heavily rely on traditional classification algorithms (Fig 7). It is worth knowing that Discriminant Analysis (DA) remains the preferred method, where the Linear Discriminant Analysis (LDA) is the most popular form in our reviewed publications. The LDA is based on Fisher's linear discriminant, which mathematically discovers a linear combination of features that maximally divide the data into two or more classes [116]. It generates linear boundaries and is regarded as a simple approach. Despite its simplicity, LDA often brought out robust and acceptable classification results for previous EVE-BCIs. Depending on various parameter settings, other deformed DAs such as Bayesian Linear Discriminant Analysis (BLDA) [1, 4, 36, 75], Shrinkage LDA [30, 31, 38, 55, 86], Regularized Discriminant Analysis (RDA)) [24, 33, 42, 43, 51, 62, 99], and step-wise Linear Discriminant Analysis (SWLDA) [74, 88, 91, 92, 98] were also used. In terms of popularity, followed by the DA, another robust supervised learning model, linear Support Vector Machine (linear SVM), was also implemented in [32, 33, 73, 82, 84, 89, 95]. The SVM mechanism is also simple: the algorithm generates a line or a hyperplane that has the largest distance to the support vectors (data points) in different categories [117]. It statistically figures out the 'best' margin to separate classes, which reduces the misclassification error on the prediction. It generally has a comparable performance with DA in practice.

In addition to the linear separation function, probability-based methods (Logistic Regression (LR) [118] and Gaussian mixture model (GMM) [119]) were also implemented in our reviewed articles. They both estimate the probability of data points that belong to a particular category. The LR is used to deal with binary classification, where the outcome variable only has two levels, and the probability distribution is pre-defined by the sigmoid function [120]. On the other hand, GMM is workable for multi-label classification. The observation in an overall population estimates its probability distribution and make the label prediction. They both gained considerable attention from the EVE-BCI community. Moreover, the simple threshold method was used in the SSSEP-based EVE-BCI. It is the simplest classification

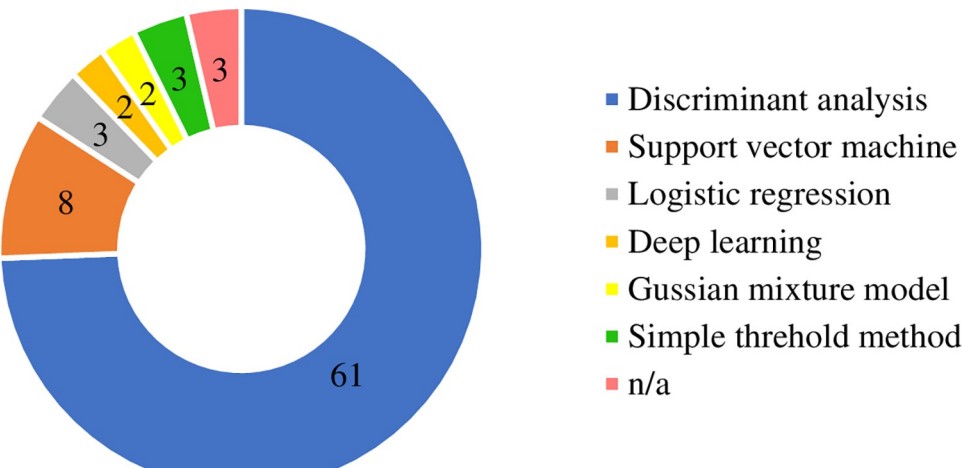

**Fig 7. Types of classifiers used in EVE-BCI development.** "n/a" refers to the paper did not report the types of classifiers.

method where pre-defined boundaries are built without any learning from the data. For instance, Han, Liu [65] proposed an approach where the dominant frequency was first extracted from the raw EEG signal using a modified FFT technique. After the extraction, the prediction was made by comparing its dominant frequency to 17, 21, and 25 Hz. The simple design is the significant advantage of this technique. However, it requires sophisticated pre-processing and feature extraction beforehand. Opposite to the simple threshold method, a deep learning (DL) approach has a far more complicated model architecture but is usually accompanied by plain pre-processing and feature generation. Thanks to the prosperity of artificial intelligence, deep learning has been a successful model in BCI field. Nevertheless, in the specific EVE-BCI sector that we reviewed, only two publications [34, 104] utilized a DL strategy. The two-dimensional EEG input (number of samples × number of channels) was fed into DL layers for learning and prediction.

## Reported performance

**Performance metrics.** The EVE-BCI is technically an EEG pattern recognition system. To be expected, most studies included performances obtained from confusion matrices, such as accuracy, true-positive rate (TPR), false-positive rate (FPR), the area under the ROC curve (AUC), and kappa value (Fig 8). In the P300 diagram, as the ratio of non-target and target was usually high (above 3:1), the kappa value and area under the ROC curve (AUC) that are more robust to the imbalanced dataset were preferable [33, 67]. In addition to maintaining good classification accuracy, classification speeding is an essential consideration for the BCI system. Thus, Information Transfer Rate (ITR) [1], which caters to both classification accuracy and speed, was also a popular measure for EVE-BCI. It is defined as the amount of information transferred per unit of time and is usually calculated in bit/mins. However, two collected publications [43, 99] do not use any measures as mentioned earlier. They probably were published by one research group in different stages of the same project. The project was about an automatic wheelchair based on the EVE-BCI system. The performance of the wheelchair was tested by going through multiple routes with obstacles. The successful rate and completion time determines the performance of the wheelchair.

**Evaluation setting.** The evaluation setting has a crucial influence on the BCI performance. As EEG signals often vary between individuals, subject-dependent and subject-independent classification have performance discrepancies. The subject-dependent classification is trained and tested by data from a single person. The evaluation in this way usually has a better

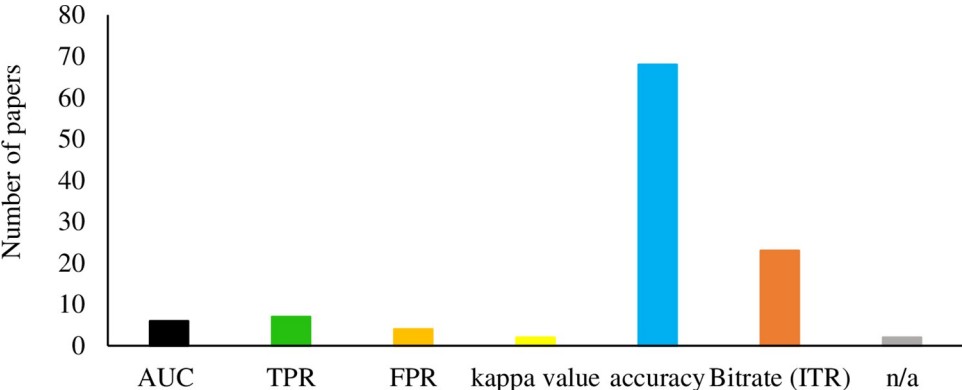

**Fig 8. Performance metrics reported in EVE-BCI development.** "n/a" refers to the paper did not report the types of performance metrics.

outcome because fewer data variabilities are required to be dealt with by the model. However, the result only shows how the model performs towards the specific subject. In other words, the model may not be generalizable when applied to other people. In the subject-independent setting, the model is trained and used on the data from multiple subjects. It leads to a more ecological scenario and is suitable for real-world application, although it requires better pattern recognition for the data with more considerable variability. Under this circumstance, the performance of the model may be relatively lower than that of the preceding one, but the result is far more objective and deliver a better estimation in the overall population. Only two reviewed articles [34, 104] carried out the subject-independent classification. DL approaches were evaluated in a cross-subject validation way in these two studies. For instance, the classifier was trained with the data from subjects 1 to 9 but tested from subject 10. This process was recursively repeated ten times, with data from each subject being used as the testing set once. Alternatively, the remaining papers either solely implemented the subject-independent setting or did not report whether they operated the subject-dependent or subject-independent classification.

Online and offline evaluation is another critical factor in BCI performance. Offline assessment is executed with the data already available. It is usually executed after the data collection to make use of the whole dataset [121]. Hence, it ignores the global timeline, and no feedback is provided to the user during the data collection. The goals of offline evaluation are to estimate the performance of the model on all the data presented in the database and figure out the best possible settings of the model up to the knowledge at the time of evaluation. It was a common method to assess the EVE-BCI, which accounts for around 80% of cases (N = 63) of our reviewed papers. Alternatively, an online evaluation is used with the non-existent data at a given time. It is designated to provide instant feedback to the user. The model evaluated online usually inherits the best setting obtained from the offline evaluation. The online assessment produces a more "realistic" performance of the EVE-BCI, as all conditions such as artifact occurrence and user distractions should be considered. Thirty-three studies carried out the online evaluation. Among 19 of those, offline evaluations were also performed.

## Feasibility of EVE-BCI

Results in Wilcoxon signed-rank tests (Fig 9) showed that accuracies produced by EVE-BCI systems were significantly higher than chance levels among different groups including healthy subjects (754.9% vs. 31.03%, N = 51, Z = -6.21, p < 0.001), patients (64.44% vs. 19.92%,

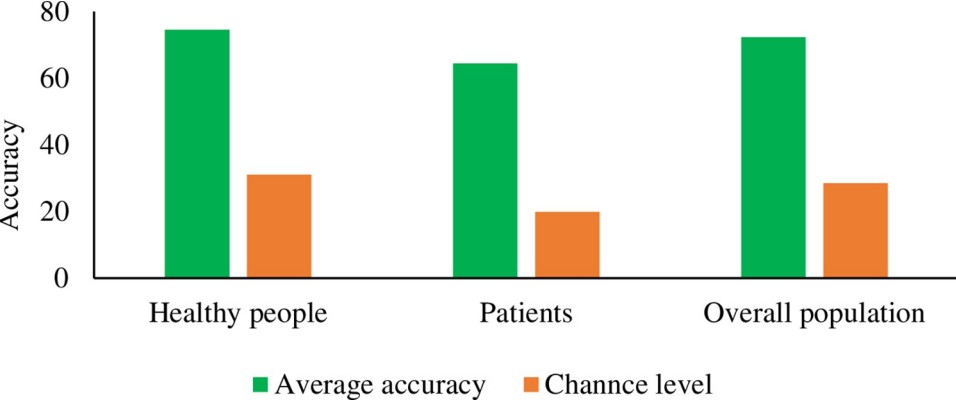

**Fig 9. Comparison between accuracies and chance levels of EVE-BCI across multiple populations.**

N = 15, Z = -3.41, p < 0.001), and the overall population (72.28% vs. 28.50% N = 66, Z = -7.06, p < 0.001). The feasibility of EVE-BCI was presented in all these population categories and, more importantly, for patients who need it most. The accuracy in each study was higher than its corresponding chance level. Four EVE-BCI systems achieved 100% accuracy. Three of these systems were found in healthy subjects and one in the patient group. However, they were all based on the offline evaluation setting. Reporting results were produced in the feature and hyperparameter optimization step. No online evaluation (real-world setting) was found to achieve the perfect accuracy.

## Discussion

According to our review, the research interest has increased significantly since the first EVE-BCI was proposed in 2006. It is a fascinating alternative for the motor imagery and visual-based BCI system. It cannot only avoid the BCI' illiteracy' issue but also be suitable for those who entirely lose the motor ability of eyeball movement. However, in terms of popularity, EVE-BCI is far behind the motor imagery and visual-based ones due to the uncertainty of its feasibility. The present review summarizes vital trends in EVE-BCI development and provides synthetic evidence on its practicability. In this section, essential elements mentioned in the result are discussed. We also deliver recommendations on the further direction of EVE-BCI and perform a checklist for a clear EVE-BCI presentation in further publications.

### Data collection

We explored two factors, including the target subject and EEG acquisition, in the data collection of EVE-BCI development. We found that patient subjects were recruited in approximately 20% of reviewed publications. By being trained and tested with these data, EVE-BCIs exhibited a good performance in practical applications. Nevertheless, only healthy subjects were included in the remaining 80% of articles. An emphasis that, EVE-BCI aims to provide an external communication channel for patients with motor ability impairment, is made in several of these studies [4, 33, 51]. However, neither training nor evaluation of their EVE-BCIs was implemented using patients' data. The external validity of their experiments is questionable. The accessibility may be the main reason for data collection from healthy subjects. As many studies were conducted in universities, researchers could conveniently access college students. Furthermore, several papers returned from our searching strategies were published in conference proceedings. Experiments in these articles were sometimes regarded as pilot studies by several researchers. They only preliminarily tested the feasibility of EVE-BCI using data from those who were most accessible [42, 43, 99]. Under such circumstances, EVE-BCI systems were usually built upon healthy subjects' data. However, the primary focus of BCI is to offer assistance for the disabled. Further studies should focus more on patient recruitment instead. This will address the existing gap and develop a reliable EVE-BCI in practice.

The essential characteristics of EEG acquisition, including acquisition form and channel number, were also reviewed. In line with expectations, most EVE-BCIs were developed by using cap-style devices. This type of device is mature and provides a stable EEG signal obtainment. However, it is worth pointing out that a unique acquisition form named fEEGrid [64] where EEG electrodes pre-attached adhesive foam stickers on the forehead. It overcomes physical pain caused by the squeeze pressure of the EEG cap during a long-time EEG collection. Hence, its analogy and itself are more suitable for long-term EEG collection scenarios, such as wheelchair and conversation speller, in contrast with the traditional cap form.

## Stimulation paradigm

P300 and SSSEP were the two most popular paradigms and accounted for predominant cases in our reviewed articles. Researchers could conveniently set up these two paradigms, and more importantly, the evoked brain activities were evident [74, 85]. In addition to using either a single P300 or SSSEP, Breitwieser, Pokorny [30] proposed a novel way to build the EVE-BCI by combining both two, where transient vibrating stimuli (for P300) were added in a constant vibration (for SSSEP). This innovative approach offered more discriminative features and increased the model's performance.

We also found that no asynchronous design, where the user can manage the BCI freely without any clues or time strains, was returned among articles reviewed in the present work. Current asynchronous BCIs are usually based on motor imagery or Steady-state visually evoked potential (SSVEP) [122]. However, they suffer respectively from issues of BCI "illiteracy" and visual impairment, as mentioned above. Several reviewed studies figured out the distinct difference in brain activities between the idle state and the concentration on persistent vibrations (SSSEP) [35, 72], which indicated that SSSEP also had great potential in acting as a brain switch [122]. This virtual switch imitates a physical switch's function to control the on (control) and off (non-control) state of the BCI system. It continuously detects the user's intention to turn on a control state from a non-control state or turn off a control state to a non-control state. The switch mechanism offers the user a way to use the BCI whenever they want and close it if they do not need it. The SSSEP application to brain-swift can be explored in further studies to investigate the feasibility of a more practical EVE-BCI, i.e., an asynchronous one.

## Vibrotactile control

Researchers in most reviewed publications did not report reasons for their choice of SOA. It seems that they solely followed their preference. As the SOA is a significant factor that influences vibration, it is crucial to justify the option in the study. The non-report of reasons for their choice of SOA maybe because only two reviewed articles [57, 91] surveyed how the SOA affected the performance of EVE-BCI. It is hard for them to explain the reason for their choices by using the proceeding evidence. We also found that the random off-time was applied to reduce the EEG collection's adaptive effect. For instance, as shown in the result, He and Contreras-Vidal [50] set the off-time as a random period between 2 to 5 seconds. The off-time is usually treated as a small resting interval when the participant's brain may be in an idle state. Therefore, randomizing the off-time may not have a crucial influence on decreasing the adaptive effect. The randomization in stimulus on-time may be an exciting alternative. The participant usually needs to execute specific experimental tasks and is in a highly focused status during this period. A random change can be more evidently recognized for breaking down the adaptive reaction.

The location of vibration was the factor most investigated by previous researchers. Non-implanted vibrations were nearly employed on every part of the human body. It was assumed that an individual had a stronger perception of the difference between multiple vibrations with an increase in the distance between them [123, 124]. Such a perception may benefit the effectiveness of EVE-BCI. Several research groups did favour the long-distance design with arranging tactors in different limbs. In contrast, it was also noted that tactors were densely attached in a specific area such as fingers, wrists, or the waist in many of our reviewed articles. This finding conveys that the distance may not be the priority in tactor position arrangement. Alternatively, the primary consideration may be the sensitivity of vibration. The places selected for tactor placement were usually fingers, wrists, and

waist. These positions are sensitive to the vibrotactile stimulus [125–127]. In line with short-distance vibration preference in EVE-BCI development, previous researchers may believe that the sensitivity has a more considerable impact on the BCI performance than the distance between tactors.

## EEG signal processing

The result shows that the pre-processing step was a popular choice for EVE-BCI development. It was operated in over 90% of cases in our reviewed publications. As the EEG is a low SNR signal, it is hard to deploy an advanced feature extraction and accurate prediction without pre-processing. The trade-off between effectiveness and automation may be the prior consideration to the option of pre-processing techniques. Most approaches used in our reviewed articles depended on mathematical computations and were automatically performed in the software [55, 60]. These methods can not only save human resources but also avoid subjective judgments. However, several artifacts caused by muscular movements cannot be removed. In response to this gap, manual artifact removal strategies were conducted after the automatic operations [47, 56]. Overall, the pre-processing stage was encouraged in the EVE-BCI development.

Various classification models were constructed for EVE-BCIs. These models can be categorized into the simple threshold, simple linear/probability, and DL approaches in terms of the model complexity. The most popular model in our reviewed publications was the simple linear one. It had an overall good performance across different populations. For instance, Guger, Spataro [58] proposed an EVE-BCI system depending on LDA that achieved both 80% accuracy for healthy subjects and ALS patients against a chance level of 12.5%. Despite the excellent performance of current models, two types of other algorithms (i.e., non-linear machine learning and deep learning models) that have been successfully applied in EEG analysis are also worthwhile exploring further. For example, in addition to performing linear classification, SVMs can efficiently perform non-linear discrimination using the kernel trick, implicitly mapping their inputs into high-dimensional feature spaces [128]. Furthermore, two systematic reviews summarized the intensive efforts in deep learning that have been made for EEG pattern recognition [129, 130]. In the EVE-BCI sector, only convolutional neural networks (one type of deep learning model) were used in two reviewed articles and achieved excellent performance [34, 104]. Other architectures, including recurrent neural networks (RNN), famous for dealing with time-series data, can also be tried. One of the advantages of DL method is to waive complex feature extractions [131]. The raw data can be directly fed into the model, where patterns can be learned without handcraft feature engineering. The second advantage of deep learning is its outstanding performance. It has been proven to achieve a state-of-the-art outcome in many fields, including computer vision [132], natural language [133], and even motor imagery BCI [129, 130]. It probably can also lead to big progress in our EVE-BCI area.

## Reported performance

For the evaluation setting, only offline and subject-dependent assessments were performed in many reviewed articles. These papers were usually used to exchange ideas at conferences and published in the proceedings. An initial evaluation in such a simple way was usually sufficient for this usage. However, both online and subject-independent evaluations should be involved to bring a comprehensive evaluation of the EVE-BCI. Online evaluation deals with more data variabilities, such as the artifact occurrence and subjects being distracted, that appear in the real world. Subject-independent assessment is defined as training and testing by the data from

multiple subjects. Hence, it can indicate the external validity and generalizability of EVE-BCI in the general population rather than a specific person. The evaluation carried out in such two ways may provide a reliable performance of an EVE-BCI in the practical environment and be encouraged in future studies.

## Feasibility of the EVE-BCI

Our review aims to investigate the feasibility of the EVE-BCI (i.e., the accuracy of prediction against the chance level). According to our statistical analyses, EVE-BCI shows a feasibility across healthy subjects, motor impairment patients, and the overall population. Although we get encouraging outcomes in statistical analyses, it is necessary to interpret these results dialectically. First, significant variations of results were presented across multiple papers. For example, researchers used a similar methodology and model in two studies [52, 72], but a 12% accuracy gap was exhibited due to different subjects and evaluation settings. Second, articles that deliver positive results (e.g., performance higher than channel level) tend to be published [134], so our analyses' significant effect may be overestimated. Although biases above may exist in our review, the gap between the overall performance of EVE-BCI and chance level was decent. Furthermore, four articles reported perfect 100% accuracy. The EVE-BCI can be conservatively considered as being feasible for the user's intention recognition based on overall and the-state-of-art performance. It may act as a powerful supplement in the BCI scientific community.

## Recommendation

The EVE-BCI system is a feasible device for detecting the user's intention. However, we recognize that several gaps had not been explored in this area. In line with these gaps, we provide the following recommendations for future studies:

- ◆ Data: recruit more motor-impaired patients; explore other types of form instead of the cap for EEG collection.

- ◆ Stimulation paradigm: test the feasibility of the asynchronous EVE-BCI design by applying the brain switch mechanism using SSSEP.

- ◆ Vibrotactile control: investigate the effectiveness of stimulus on-time randomization on the reduction of adaptive effect.

- ◆ EEG signal processing: explore the non-linear machine learning and deep learning approaches.

- ◆ Reported performance: present subject-independent training and online evaluation.

Throughout reading published papers in EVE-BCI, we identify that several works may be presented in an unclear way. To ensure the quality and avoid the ambiguity of work presentation, we proposed the following checklist (Table 6) for essential elements of an EVE-BCI that should be reported:

## Conclusions

Brain-computer interfaces are going through technological progress, and the vibrotactile modality is drawing increasing focus from researchers in the BCI community. However, compared to other types of mature BCI, such as motor imagery and visual-based ones, the EVE-BCI remains at the initial stage, with various challenges still waiting to be tackled. In this review, we summarises the development status of EVE-BCI by analysing 79 studies from January 2006 to December 2021.

**Table 6. Recommendations on key elements in the future study.**

| Recommendation | Key elements |
|---|---|
| Data collection | Clearly describe:<br>❖ type of subject<br>❖ number of subjects<br>❖ type of device<br>❖ electrode montage<br>❖ number of channels (location)<br>❖ sample rate. |
| Stimulation paradigm | Clear describe:<br>❖ type of paradigm<br>❖ executions of each trial<br>❖ entire experimental produce |
| Vibrotactile control | Clear describe and justify reasons for the choices, if any:<br>❖ vibration location<br>❖ vibration frequency and intensity<br>❖ vibration number<br>❖ on-time<br>❖ off-time<br>❖ type of tactors |
| EEG signal processing | Clear describe and justify reasons for the choices, if any:<br>❖ type of filter<br>❖ artifact removal techniques (equations, if any)<br>❖ based-line correction (equations if any)<br>❖ feature extraction (equations if any)<br>❖ classification model (equations if any) |
| Reported Performance | Clearly describe:<br>❖ performance metrics (equations if any)<br>❖ evaluation setting (offline or online, cross-validating or training-testing, and subject-independent or subject-dependent) |

The following significant trends have been identified in our review: (1) the EEG data for EVE-BCI development were usually recorded from healthy subjects by a cap-style device; (2) P300 and SSSEP were the two most frequently employed paradigms in this field; (3) locations of the vibration have been heavily investigated in nearly every part of the body, whereas other vibration factors may lack attention from researchers; (4) signal pre-processing were usually carried out before the feature extraction where the temporal characteristics were usually derived and fed into a linear classification model. (5) The EVE-BCI aggregated accuracies were significantly higher than chance levels among healthy subjects, motor-impaired patients, and the overall population.

In addition to these major trends, several limitations of the current development in EVE-BCI were also introduced. This review showed more targeted works needed exploring to enhance the quality of EVE-BCI. Such works can be done in ways of data recording from motor-impaired patients, expanding other forms of EEG collection rather than using the cap style, applying the SSSEP in the brain switch mechanism for asynchronous EVE-BCI design, randomizing the stimulus on-time to decrease the adaptive effect, exploring non-linear traditional machine learning algorithms and deep learning models, and evaluating the EVE-BCI by subject-independent and online assessment. Finally, we provide recommendations on significant elements that should be presented to clearly illustrate the EVE-BCI in an article.

# Supporting information

**S1 Checklist. Preferred Reporting Items for Systematic Reviews and Meta-Analyses (PRISMA).**
(DOC)

**S1 Appendix. Searching queries.**
(DOCX)

**S1 Database. Data extracted from the included studies.** Data in this file can be used to fully replicate the results and findings presented in our study.
(XLSX)

## Author Contributions

**Conceptualization:** Xiuyu Huang, Cynthia Yuen Yi Lai, Kup-Sze Choi.

**Data curation:** Xiuyu Huang, Shuang Liang, Zengguang Li, Kup-Sze Choi.

**Formal analysis:** Xiuyu Huang, Shuang Liang, Zengguang Li, Cynthia Yuen Yi Lai, Kup-Sze Choi.

**Investigation:** Xiuyu Huang.

**Methodology:** Xiuyu Huang.

**Software:** Xiuyu Huang.

**Supervision:** Kup-Sze Choi.

**Validation:** Kup-Sze Choi.

**Writing – original draft:** Xiuyu Huang, Kup-Sze Choi.

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
