## [Decision Letter · Decision Letter 0]

30 Mar 2022

PONE-D-21-13885EEG-based Vibrotactile Evoked Brain-Computer Interfaces System: A Systematic ReviewPLOS ONE

Dear Dr. HUANG,

Thank you for submitting your manuscript to PLOS ONE. After careful consideration, we feel that it has merit but does not fully meet PLOS ONE’s publication criteria as it currently stands. Therefore, we invite you to submit a revised version of the manuscript that addresses the points raised during the review process. Please submit your revised manuscript by May 14, 2022. If you will need more time than this to complete your revisions, please reply to this message or contact the journal office at plosone@plos.org. Please include the following items when submitting your revised manuscript:A rebuttal letter that responds to each point raised by the academic editor and reviewer(s). You should upload this letter as a separate file labeled 'Response to Reviewers'.A marked-up copy of your manuscript that highlights changes made to the original version. You should upload this as a separate file labeled 'Revised Manuscript with Track Changes'.An unmarked version of your revised paper without tracked changes. You should upload this as a separate file labeled 'Manuscript'.

We look forward to receiving your revised manuscript.

Kind regards,

Sheng Ge

Academic Editor

PLOS ONE

Journal Requirements:

3. Your abstract cannot contain citations. Please only include citations in the body text of the manuscript, and ensure that they remain in ascending numerical order on first mention.

4. We note that Figure 3 in your submission contain map images which may be copyrighted. All PLOS content is published under the Creative Commons Attribution License (CC BY 4.0), which means that the manuscript, images, and Supporting Information files will be freely available online, and any third party is permitted to access, download, copy, distribute, and use these materials in any way, even commercially, with proper attribution. For these reasons, we cannot publish previously copyrighted maps or satellite images created using proprietary data, such as Google software (Google Maps, Street View, and Earth). For more information, see our copyright guidelines: http://journals.plos.org/plosone/s/licenses-and-copyright.

a. You may seek permission from the original copyright holder of Figure 3 to publish the content specifically under the CC BY 4.0 license.  

5. Please include a copy of Table 3 which you refer to in your text on page 8.

6. We note that this manuscript is a systematic review or meta-analysis; our author guidelines therefore require that you use PRISMA guidance to help improve reporting quality of this type of study. Please upload copies of the completed PRISMA checklist as Supporting Information with a file name “PRISMA checklist”.

Reviewers' comments:

Reviewer's Responses to Questions

**Comments to the Author**

1. Is the manuscript technically sound, and do the data support the conclusions?

Reviewer #1: Yes

Reviewer #2: Yes

2. Has the statistical analysis been performed appropriately and rigorously? 

Reviewer #1: Yes

Reviewer #2: Yes

3. Have the authors made all data underlying the findings in their manuscript fully available?

Reviewer #1: Yes

Reviewer #2: Yes

4. Is the manuscript presented in an intelligible fashion and written in standard English?

Reviewer #1: Yes

Reviewer #2: Yes

5. Review Comments to the Author

Reviewer #1: The current study presents a systematic review for EEG-based vibrotactile evoked brain-computer interface (EVE-BCI). The authors strictly follow the classical way to perform the review. To my best knowledge, such a review on this type of BCI is absent in the current stage. The authors give a comprehensive summarization of the current trend of the development in EVE-BCI. They also offer several interesting directions in the future study in this area. The manuscript might be a useful starting point for novice readers who want to commit to EVE-BCI research. However, several revisions need to be done to improve the manuscript. Detailed comments are given below.

Major comments:

1. The subsections 1.1 and 1.2 can be written more concisely, as they are not the primary focus of this review. I understand the authors may want to introduce the general brain-computer interface in detail. However, a concise context should be better.

2. Figure 1 may present that the EEG technique is applied to capture the data from participants. The current form should lead to ambiguity.

3. The publication time of studies included in this review is only till end of 2020. However, several new studies have been published recently. The authors should also further include them in the revision. I only list one example, i.e., [1]. The authors should carry out further research to include others.

[1] Eidel, M., Tröger, W., Winterholler, M., Giesler, T., & Kübler, A. (2021, February). A Tactile Brain-Computer Interface for Virtual Wheelchair Control at Home. In 2021 9th International Winter Conference on Brain-Computer Interface (BCI) (pp. 1-3). IEEE.

4. The authors may consider rewriting the last sentence of subsection 2.3. It is confusing how the studies count twice in the statistical analysis.

5. In subsection 3.3, the authors may introduce the concept of vibrotactile induced Sensory-Motor Rhythms (VT-SMR) more clearly.

6. In subsection 3.5.3, how can LDA analysis be comprehensive for the classification problem. The authors may need a further explanation.

7. The authors may consider rewriting the last three sentences of subsection 4.1. What is the meaning of “theoretically estimate the difference in EVE-BCI performance between healthy people and patients.”

8. The authors should pay attention to the correct present and past tense usage. Several mistakes regarding tense have been made in the manuscript.

Minor comments:

1. Define the full name before the abbreviation, i.e., EEG, EVE-BCI, in the abstract.

2. Subsection 1.5 may not include the context of section 1.

3. In the first line of Section 2, “Systematic review” should be “systematic” review.

4. In Figure 3, the subfigure (a) title should not contain distribution.

5. In subsection 3.2.2, what is cap shape?

6. In subsection 3.4.2, define the bitrate.

7. In subsection 4.3., non-reporting?

8. Please read through the manuscript again to correct other grammatic errors.

Reviewer #2: This study systematically reviewed published EVE-BCI studies, summarized the current trends, and analyzed the feasibility of EVE-BCIs. In addition, some practical recommendations were also provided. Overall, the paper is well-organized, and the literature survey adequately reflects the state-of-the-art. The following comments are provided to help improve the manuscript before publication.

Comment 1

The Abstract should not exceed 300 words. The methods and results should be reported briefly, while the significance of this study needs to be clarified more clearly. In addition, the reviewer suggests changing the “Background” into “Objective”.

Comment 2

Abstract: “Recently, a novel type of EEG-based BCI using the vibrotactile stimulus shows a great potential to act as an excellent alternative to other common BCIs such as motor imagery and visual-based ones.”

In Sec. 1.2 “Advantages of EEG and vibrotactile stimuli on BCI”, the author introduced the limitations of the VEP (visual-evoked potential)-BCI. However, the comparison between MI-BCI and EVE-BCI seems to be missing.

Comment 3

Sec. 1.2: “As the combination of EEG technique and vibrotactile evoked stimulus, the EVE-BCI inherits both their advantage. Its components are generally illustrated in Figure 1. Analogous to other evoked BCI systems, external vibrations are first applied to users to elicit distinct brain waves that can be captured by EEG techniques. Then, the signals are processed by various techniques for feature extraction. Finally, a classification was made for the recognition of the user’s intention as an execution command to the outside world.”

The introduction of EVE-BCI was so brief that it may be difficult for readers to understand how it works. It is necessary to introduce each component of EVE-BCI more detailly.

Comment 4

The image quality of the figures in this manuscript should be improved. And all figures should be rechecked by the authors carefully. For instance, in Fig. 3.B, Australia was missing, and Taiwan (China) was mislabeled.

Comment 5

In Sec. 3.2.2 “EEG acquisition”, the authors reviewed the EEG cap and the number of electrodes used in EVE-BCI studies. The definition of regions of interest (ROIs) in these studies should also be reviewed.

Comment 6

In Sec. 3.4 “Vibrotactile control”, the authors reviewed four essential factors: frequency, location, number, and SOA of vibrations. The reviewer wonders if vibration intensity is also an important factor or how to control vibration intensity.

Comment 7

Sec. 3.5.3: “In general, the BCI is thought as a brain signal classification problem.”

The BCI should not be narrowly defined as a classification problem. In sleep or cognitive monitoring applications, the prediction of the human state was usually obtained by regression methods. Please justify.

Comment 8

Another concern of the reviewer is the English usage. There are a few mistakes, for instance:

In Sec. 1.1, “Functional magnetic resonance imaging (fMRI)” should be “functional magnetic resonance imaging (fMRI)”.

Table 5: “spatially whiten” should be “spatial whitening”.

Please check for similar mistakes in the current manuscript and correct them.

6. PLOS authors have the option to publish the peer review history of their article (what does this mean?). If published, this will include your full peer review and any attached files.

Reviewer #1: **Yes: **Yuanpeng Zhang

Reviewer #2: **Yes: **Yi-chuan Jiang

---

## [Author Response · Author response to Decision Letter 0]

17 Apr 2022

A. Responses to Reviewer #1’s comments:

Major comments:

1. The subsections 1.1 and 1.2 can be written more concisely, as they are not the primary focus of this review. I understand the authors may want to introduce the general brain-computer interface in detail. However, a concise context should be better.

Response: 

We have removed and rewritten several unnecessary sentences to make these two subsections more concise. The removed sentences include:

It increasingly draws attention from academic and industrial areas because of its practical prospective applications.

For example, scar tissue caused by operations may reduce the brain signal's sensitivity; the brain may have the immune effect on the implanted electrodes. Thus, fewer studies are focused on the invasive BCI due to the complexity and uncertainty.

The data collection can be achieved using an EEG cap, which is regarded as being more convenient and at a lower cost than other non-invasive acquisition methods, especially to the fMRI

It is thus highly recommended to be used in a device that requires simultaneous neuro-responses.

You may check the corresponding parts in the revised manuscript for more details.

2. Figure 1 may present that the EEG technique is applied to capture the data from participants. The current form should lead to ambiguity.

Response: 

We amended the Figure 1 in the revised manuscript. You can refer it for more details. It should be clearer.

3. The publication time of studies included in this review is only till end of 2020. However, several new studies have been published recently. The authors should also further include them in the revision. I only list one example, i.e., [1]. The authors should carry out further research to include others.

[1] Eidel, M., Tröger, W., Winterholler, M., Giesler, T., & Kübler, A. (2021, February). A Tactile Brain-Computer Interface for Virtual Wheelchair Control at Home. In 2021 9th International Winter Conference on Brain-Computer Interface (BCI) (pp. 1-3). IEEE.

Response: 

We included the publications in 2021 and 2022 in the revised manuscript. A total of 163 more, including duplicates returned by the search queries, and six more were finally included in the review. This results in a total of 79 eligible papers in the revised manuscript.

4. The authors may consider rewriting the last sentence of subsection 2.3. It is confusing how the studies count twice in the statistical analysis.

Response: 

We modified the sentences as follows. It should be clearer.

For the overall population analysis, the performances of EVE-BCI for healthy subjects and patients are both considered. Several studies may contain different EVE-BCI experiments for healthy subjects and patients, respectively. They were counted two times (one for healthy and the other for patients) in the statistical test for the overall population.

5. In subsection 3.3, the authors may introduce the concept of vibrotactile induced Sensory-Motor Rhythms (VT-SMR) more clearly.

Response: 

Following your comment, we add a clear statement as follows in the revised manuscript for a clear definition to VT-SMR.

More specifically, VT-SMR is regarded as an oscillatory idle of synchronized electric brain events aroused by constant vibrotactile stimuli.

6. In subsection 3.5.3, how can LDA analysis be comprehensive for the classification problem. The authors may need a further explanation.

Response: 

For the classification problem, LDA can offer robust and good performance. The "comprehensive" should be misused in the initial submission. We have removed it in the revised manuscript.

7. The authors may consider rewriting the last three sentences of subsection 4.1. What is the meaning of “theoretically estimate the difference in EVE-BCI performance between healthy people and patients.”

Response: 

We have amended the corresponding context to better clarify this part of discussion about more recruitment on patients to develop EVE-BCI in future studies. The modified context is as below.

Furthermore, several papers returned from our searching strategies were published in conference proceedings. Experiments in these articles were sometimes regarded as pilot studies by several researchers. They only preliminarily tested the feasibility of EVE-BCI using data from those who were most accessible. Under such circumstances, EVE-BCI systems were usually built upon healthy subjects' data. However, the primary focus of BCI is to offer assistance for the disabled. Further studies should focus more on patient recruitment instead. This will address the existing gap and develop a reliable EVE-BCI in practice.

8. The authors should pay attention to the correct present and past tense usage. Several mistakes regarding tense have been made in the manuscript.

Response: 

We have corrected the tense mistakes in the revised manuscript.

Minor comments:

1. Define the full name before the abbreviation, i.e., EEG, EVE-BCI, in the abstract.

2. Subsection 1.5 may not include the context of section 1.

3. In the first line of Section 2, “Systematic review” should be “systematic” review.

4. In Figure 3, the subfigure (a) title should not contain distribution.

5. In subsection 3.2.2, what is cap shape?

6. In subsection 3.4.2, define the bitrate.

7. In subsection 4.3., non-reporting?

8. Please read through the manuscript again to correct other grammatic errors.

Response: 

1. We have defined all abbreviations in the abstract.

2. We have removed the corresponding context.

3. We have changed "Systematic review" into "systematic review".

4. We have removed the word “distribution”.

5. We have changed the “cap shape” as “cap style”.

6. We have defined the bitrate as bits per minute.

7. We have amended the "non-reporting" as "non-report of reasons for their choice of SOA".

8. We have double checked the manuscript and corrected all grammatic errors.

B. Responses to Reviewer #2’s comments:

Comment 1

The Abstract should not exceed 300 words. The methods and results should be reported briefly, while the significance of this study needs to be clarified more clearly. In addition, the reviewer suggests changing the “Background” into “Objective”.

Response: 

*1. We shortened the Abstract to less than 300 words. As suggested, we reported methods and results more briefly in the revised manuscript as follows.

Method

Five major databases were searched for relevant publications. Multiple key concepts of EVE-BCI, including data collection, stimulation paradigm, vibrotactile control, EEG signal processing, and reported performance, were derived from each eligible article. We then analyzed these concepts to reach our objective.

Results

(a) seventy-nine studies are eligible for inclusion; (b) EEG data are mostly collected among healthy people with an embodiment of EEG cap in EVE-BCI development; (c) P300 and Steady-State Somatosensory Evoked Potential are the two most popular paradigms; (d) only locations of vibration are heavily explored by previous researchers, while other vibrating factors draw little interest. (e) temporal features of EEG signal are usually extracted and used as the input to linear predictive models for EVE-BCI setup; (f) subject-dependent and offline evaluations remain popular assessments of EVE-BCI performance; (g) accuracies of EVE-BCI are significantly higher than chance levels among different populations. 

*2. We also better clarified the significance of the present study with more descriptions as follows.

Significance

we summarize trends and gaps in the current EVE-BCI by identifying influential factors. A comprehensive overview of EVE-BCI can be quickly gained by reading this review. We also provide recommendations for the EVE-BCI design and formulate a checklist for a clear presentation of the research work. They are useful references for researchers to develop a more sophisticated and practical EVE-BCI in future studies.

*3. We also changed the “Background” into “Objective”.

Comment 2

Abstract: “Recently, a novel type of EEG-based BCI using the vibrotactile stimulus shows a great potential to act as an excellent alternative to other common BCIs such as motor imagery and visual-based ones.”

In Sec. 1.2 “Advantages of EEG and vibrotactile stimuli on BCI”, the author introduced the limitations of the VEP (visual-evoked potential)-BCI. However, the comparison between MI-BCI and EVE-BCI seems to be missing.

Response: 

Yes, it is missing. According to your suggestion, we also presented the limitations of MI-BCI in the revised manuscript as below. The P300 EVE-BCI can address such limitations, and we mentioned it in the revised manuscript. 

Among EEG-based BCIs, the motor imagery (MI) paradigm is one of the most common options. It leads to an event-related desynchronization (ERD) which can be used to detect the user’s intention. However, it has two major drawbacks highlighted by researchers. First, motor/movement imagination is an active mental process. Several people may not be able to execute this task. The other one is that ERD patterns of MI require a relatively long time (several seconds) to appear, which limits the practical usage in real-time BCIs. 

Comment 3

Sec. 1.2: “As the combination of EEG technique and vibrotactile evoked stimulus, the EVE-BCI inherits both their advantage. Its components are generally illustrated in Figure 1. Analogous to other evoked BCI systems, external vibrations are first applied to users to elicit distinct brain waves that can be captured by EEG techniques. Then, the signals are processed by various techniques for feature extraction. Finally, a classification was made for the recognition of the user’s intention as an execution command to the outside world.”

The introduction of EVE-BCI was so brief that it may be difficult for readers to understand how it works. It is necessary to introduce each component of EVE-BCI more detailly.

Response: 

Yes, we have amended this paragraph accordingly. A concrete example of EVE-BCI, including its illustrating figure, is also presented for a clear explanation of EVE-BCI. The modified paragraph in the revised manuscript is given below. You may check the modified figure in the revised manuscript.

As a combination of EEG technique and vibrotactile evoked stimulus, EVE-BCI inherits both their advantages. Its components are generally illustrated in Fig 1 (top). Analogous to other evoked BCI systems, external vibrations are first applied to users to elicit distinct brain waves that EEG techniques can capture. Then, the signals are processed by various techniques for feature extraction. Finally, a classifier recognizes the user’s intention as an execution command to the outside world. A concrete example of EVE-BCI using the P300 paradigm is shown in Fig 1 (bottom). Two vibrators are attached to the subject’s left and right index fingers. The subject is asked to pay attention to either finger. Let’s assume that he/she focuses on the vibration on the left (target). Left and right tactors keep vibrating in random designated order (the right stimulator vibrates four times as many as the left one at the training phase; two vibrators have the same vibration times at the evaluation phase). An EEG cap records the subject’s corresponding brain signals while tactors are vibrating. These signals contain distinct patterns and can be converted into feature vectors by a signal processing method called common spatial pattern (CSP). The logistic regression algorithm predicts the user’s focus based on mathematical calculations. It is noticed that there are many settings (e.g., locations of vibrator, signal processing method, and predictive algorithm) in such a simple EVE-BCI. Other choices may contribute to a better design. For example, we can apply the vibrators to wrists instead of fingers. Interesting variables that may advance the performance of EVE-BCI are significant parts of our review.

Comment 4

The image quality of the figures in this manuscript should be improved. And all figures should be rechecked by the authors carefully. For instance, in Fig. 3.B, Australia was missing, and Taiwan (China) was mislabeled.

Response: 

We are sincerely unaware of the mistakes in Fig. 3. (b). We have amended this figure as a bar chart. You may check it in the revised manuscript. We also carefully check other figures and improve their resolutions and illustration clarity.

Comment 5

In Sec. 3.2.2 “EEG acquisition”, the authors reviewed the EEG cap and the number of electrodes used in EVE-BCI studies. The definition of regions of interest (ROIs) in these studies should also be reviewed.

Response: 

Yes, regions of interest (ROIs) are defined and analyzed in some EEG studies [1,2,3]. However, to my best knowledge, the reviewed EVE-BCI studies do not define regions of interest (ROIs) in their experiments. The researchers instead have significant interests in the sensorimotor cortex region in EVE-BCI development, according to our review. Therefore, we included a paragraph blow in the revised manuscript to summarize our corresponding findings. 

[1] Mehta, K., & Kliewer, J. (2015). An information theoretic approach toward assessing perceptual audio quality using EEG. IEEE Transactions on Molecular, Biological and Multi-Scale Communications, 1(2), 176-187.

[2] Mitchell, M. B., Shirk, S. D., McLaren, D. G., Dodd, J. S., Ezzati, A., Ally, B. A., & Atri, A. (2016). Recognition of faces and names: multimodal physiological correlates of memory and executive function. Brain imaging and behavior, 10(2), 408-423.

[3] Aung, K. P. P., & Nwe, K. H. (2020, November). Regions of Interest (ROI) Analysis for Upper Limbs EEG Neuroimaging Schemes. In 2020 International Conference on Advanced Information Technologies (ICAIT) (pp. 53-58). IEEE.

Articles in the review suggest that the most frequently used number of channels ranged from 3 to 8. Among these channels, C3, Cz and C4 draw significant interest from researchers. They cover the primary sensorimotor cortex sensitive to somatosensory stimulation and record corresponding neural responses. The first EVE-BCI was developed using EEG data collected in this region. Since then, over 92% (73 out of 79) of studies used the data from C3, Cz and C4 in their BCI experiments. The rest, only a tiny portion, either only used data from other channels or did not report any relevant details. The patterns of brain events were also visualized in reviewed articles. Twenty-eight reviewed articles selected specific channels, instead of all, for the visualization. More than 85% of these publications chose at least either one of the C3, Cz and C4 channels to present the patterns. Those results suggest the intensive interest of the primary sensorimotor cortex region in the EVE-BCI development.

Comment 6

In Sec. 3.4 “Vibrotactile control”, the authors reviewed four essential factors: frequency, location, number, and SOA of vibrations. The reviewer wonders if vibration intensity is also an important factor or how to control vibration intensity.

Response: 

That’s an interesting point. We showed the data on vibration intensity in the revised manuscript. The details can be checked as follows. As not many studies reported the intensity of the vibration, we also recommend researchers report it (i.e., we added it to the checklist in the Discussion section of the revised manuscript) in future studies to ensure the reproducibility of their experiments.

In addition to the frequency, the intensity of vibrators is also explored in this review. Only around 16% (n=13) of articles reported their choices of intensity ranging from 0.6-4.0 G. They normally stated that participants have explicit feelings about the vibration, but no special or sophisticated designs were applied. Furthermore, most reviewed publications do not present any details about the vibrotactile intensity.

Comment 7

Sec. 3.5.3: “In general, the BCI is thought as a brain signal classification problem.”

The BCI should not be narrowly defined as a classification problem. In sleep or cognitive monitoring applications, the prediction of the human state was usually obtained by regression methods. Please justify.

Response: 

Yes, you are right. BCI is not solely defined as a classification device. In EVE-BCI, all articles reviewed are classification-based. We amended this sentence as below for a more precise expression in the revised manuscript according to your suggestion.

In general, EVE-BCI in reviewed articles focused on classifying the user’s intention using distinct patterns of certain brain events.

Comment 8

Another concern of the reviewer is the English usage. There are a few mistakes, for instance:

In Sec. 1.1, “Functional magnetic resonance imaging (fMRI)” should be “functional magnetic resonance imaging (fMRI)”.

Table 5: “spatially whiten” should be “spatial whitening”.

Please check for similar mistakes in the current manuscript and correct them.

Response: 

Thanks for pointing them out. We double-checked the grammatic mistakes in the initial manuscript and corrected them all in the revised submission.

---

## [Decision Letter · Decision Letter 1]

13 May 2022

EEG-based vibrotactile evoked brain-computer interfaces system: a systematic review

PONE-D-21-13885R1

Dear Dr. HUANG,

We’re pleased to inform you that your manuscript has been judged scientifically suitable for publication and will be formally accepted for publication once it meets all outstanding technical requirements.

Kind regards,

Sheng Ge

Academic Editor

PLOS ONE

Reviewers' comments:

Reviewer's Responses to Questions

**Comments to the Author**

1. If the authors have adequately addressed your comments raised in a previous round of review and you feel that this manuscript is now acceptable for publication, you may indicate that here to bypass the “Comments to the Author” section, enter your conflict of interest statement in the “Confidential to Editor” section, and submit your "Accept" recommendation.

Reviewer #1: All comments have been addressed

Reviewer #2: All comments have been addressed

2. Is the manuscript technically sound, and do the data support the conclusions?

Reviewer #1: Yes

Reviewer #2: Yes

3. Has the statistical analysis been performed appropriately and rigorously? 

Reviewer #1: Yes

Reviewer #2: Yes

4. Have the authors made all data underlying the findings in their manuscript fully available?

Reviewer #1: Yes

Reviewer #2: Yes

5. Is the manuscript presented in an intelligible fashion and written in standard English?

Reviewer #1: Yes

Reviewer #2: Yes

6. Review Comments to the Author

Reviewer #1: The authors made a compenhensive regarding EEG-based vibrotactile evoked brain-computer interfaces system. In the revised manuscript, the authors addressed my issues carefully. Now I have no new comments.

Reviewer #2: This study systematically reviewed published EVE-BCI studies, summarized the current trends, and analyzed the feasibility of EVE-BCIs. In addition, some practical recommendations were also provided. Overall, the paper is well-organized, and the literature survey adequately reflects the state-of-the-art. I recommend this article to be published in PLOS ONE.

7. PLOS authors have the option to publish the peer review history of their article (what does this mean?). If published, this will include your full peer review and any attached files.

Reviewer #1: No

Reviewer #2: **Yes: **Yichuan Jiang

---

## [Editor Report · Acceptance letter]

27 May 2022

PONE-D-21-13885R1 

EEG-based vibrotactile evoked brain-computer interfaces system: a systematic review 

Dear Dr. Huang:

I'm pleased to inform you that your manuscript has been deemed suitable for publication in PLOS ONE. Congratulations! Your manuscript is now with our production department. 

Kind regards, 

on behalf of

Dr. Sheng Ge 

Academic Editor

PLOS ONE